# ON THE EFFECTIVENESS OF DISCRETE REPRESENTATIONS IN SPARSE MIXTURE OF EXPERTS

## ABSTRACT

Sparse mixture of experts (SMoE) is an effective solution for scaling up model capacity without increasing the computational costs. A crucial component of SMoE is the router, responsible for directing the input to relevant experts; however, it also presents a major weakness, leading to routing inconsistencies and representation collapse issues. Instead of fixing the router like previous works, we propose an alternative that assigns experts to input via *indirection*, which employs the discrete representation of input that points to the expert. The discrete representations are learnt via vector quantization, resulting in a new architecture dubbed Vector-Quantized Mixture of Experts (VQMoE). We provide theoretical support and empirical evidence demonstrating the VQMoE's ability to overcome the challenges present in traditional routers. Through extensive evaluations on both large language models and vision tasks for pre-training and fine-tuning, we show that VQMoE achieves a 28% improvement in robustness compared to other SMoE routing methods, while maintaining strong performance in fine-tuning tasks.

## 1 INTRODUCTION

Scaling Transformers with data and compute has demonstrated unprecedented successes across various domains such as natural language processing (NLP) tasks (Du et al., 2022; Fedus et al., 2022; Zhou et al., 2024), and visual representation learning (Riquelme et al., 2021a; Shen et al., 2023b). However, training and inference of a single large Transformer-based model might require hundreds of thousands of compute hours, costing millions of dollars (Kaddour et al., 2023). This issue has motivated contemporary studies to investigate Sparse Mixture-of-Experts (SMoE) (Shazeer et al., 2017; Zoph et al., 2022; Xue et al., 2024; Jiang et al., 2024). SMoE models that are inspired by (Jacobs et al., 1991a) usually include a set of experts sharing the same architecture and a router that activates only one or a few experts for each input. Compared to dense models of the same size, SMoE counterparts significantly reduce inference time thanks to not using all experts simultaneously (Artetxe et al., 2022; Krajewski et al., 2024).

However, training SMoEs remains a challenge due to representation collapse, that is, either a small number of experts receive most of the routed tokens or all experts converge to learn similar representations. To tackle the issue, several works (Chi et al., 2022; Chen et al., 2023a; Do et al., 2023) have focused on router policy improvement. However, these do not touch a fundamental question, 'Do we really need a router in the first place?' Our research suggests that adopting a discrete representation could help solve the challenges currently faced by the router method. Discrete representation learning in the context of SMoE is motivated by its ability to capture structured and interpretable patterns within data, aligning with the way that humans categorize and process information through distinct symbols, like tokens. This approach enables better generalization and facilitates knowledge transfer across different contexts. Additionally, discrete representations provide a robust and efficient mechanism for selecting and routing inputs to the appropriate experts by clustering them more effectively. By bridging the gap between discrete and continuous representations, this method leads to more stable and interpretable expert assignments, helping to mitigate issues such as representation collapse and overfitting, which are common challenges in SMoE training.

Employing vector quantization (VQ) techniques to learn discrete representation, this paper proposes a novel mixture of expert framework, named VQMoE, which overcomes the representation collapse and inconsistency in training sparse mixture of experts. More specifically, we prove that the existing

router methods are inconsistent and VQMoE suggests an optimal expert selection for training SMoE. Additionally, our method guarantees superior SMoE training strategies compared to the existing methods by solving the representation collapse by design.

We evaluate the proposed method by conducting pre-training of Large Language Models (LLMs) on several advanced SMoE architectures, such as SMoE (Jiang et al., 2024), StableMoE (Dai et al., 2022), or XMoE (Chi et al., 2022), followed by fine-tuning on downstream tasks on both Language and Vision domains.

In summary, the primary contributions of this paper are threefold: (1) we theoretically demonstrate that learning a discrete representation is an optimal approach for expert selection and that VQMoE inherently addresses the issue of representation collapse; (2) we propose the use of the Vector Quantization method to learn cluster structures and resolve related challenges; and (3) we conduct extensive experiments on large language models and vision pre-training and fine-tuning tasks, providing an in-depth analysis of VQMoE's behavior to showcase its effectiveness.

## 2   RELATED WORK

**Sparse Mixture of Experts (SMoE).** Sparse Mixture of Experts (SMoE) builds on the Mixture of Experts (MoE) framework introduced by Jacobs et al. (1991b); Jordan & Jacobs (1994), with the core idea that only a subset of parameters is utilized to process each example. This approach was first popularized by Shazeer et al. (2017). SMoE's popularity surged when it was combined with large language models based on Transformers (Zhou et al., 2022; Li et al., 2022; Shen et al., 2023a), and its success in natural language processing led to its application across various fields, such as computer vision (Riquelme et al., 2021b; Hwang et al., 2023; Lin et al., 2024), speech recognition (Wang et al., 2023; Kwon & Chung, 2023), and multi-task learning (Ye & Xu, 2023; Chen et al., 2023b).

However, SMoE faces a major problem in training known as representation collapse, i.e., the experts converge to similar outputs. To address this, various methods have been introduced. XMoE (Chi et al., 2022) calculates routing scores between tokens and experts on a low-dimensional hypersphere. SMoE-dropout (Chen et al., 2023a) uses a fixed, randomly initialized router network to activate experts and gradually increase the number of experts involved to mitigate collapse. Similarly, HyperRouter (Do et al., 2023) utilizes HyperNetworks (Ha et al., 2016) to generate router weights, providing another pathway for training SMoE effectively. StableMoE (Dai et al., 2022) introduces a balanced routing approach where a lightweight router, decoupled from the backbone model, is distilled to manage token-to-expert assignments. The StableMoE strategy ensures stable routing by freezing the assignments during training, while SimSMoE Do et al. (2024) forces experts to learn dissimilar representations. Despite these extensive efforts, the representation collapse issue persists, as highlighted by Pham et al. (2024). While most solutions focus on improving routing algorithms, our approach takes a different path by learning a discrete representation of input that points to relevant experts.

**Discrete Representation.**  Discrete representations align well with human thought processes; for example, language can be understood as a series of distinct symbols. Nevertheless, the use of discrete variables in deep learning has proven challenging, as evidenced by the widespread preference for continuous latent variables in most current research. VQVAE (van den Oord et al., 2017) implements discrete representation in Variational AutoEncoder (VAE) (Kingma & Welling, 2022) using vector quantisation (VQ). IMSAT (Hu et al., 2017) attains a discrete representation by maximizing the information-theoretic dependency between data and their predicted discrete representations. Recent works follow up the vector quantisation ideas and make some enhancements for VAE, for example: (Yu et al., 2022); (Mentzer et al., 2023); and (Yang et al., 2023). Mao et al. (2022) utilize a discrete representation to strengthen Vision Transformer (ViT) (Dosovitskiy et al., 2021). To the best of our knowledge, our paper is the first to learn a discrete representation of Sparse Mixture of Experts.

## 3   METHOD

We propose a novel model, Vector-Quantized Mixture of Experts (VQMoE), which learns discrete representations for expert selection. As illustrated in Fig. 1a, our approach selects experts directly based on the input representation, eliminating the need for a trained router. To prevent information loss, we integrate discrete and continuous representations within the model.

### 3.1 PRELIMINARIES

**Sparse Mixture of Experts.** Sparse Mixture of Experts (SMoE) is often a transformer architecture that replaces the MLP layers in standard transformers with Mixture of Experts (MoE) layers (Shazeer et al., 2017). Given $\mathbf{x} \in \mathbb{R}^{n \times d}$ as the output of the multi-head attentions (MHA), the output of SMoE with $N$ experts is a weighted sum of each expert's computation $E_i(x)$ by the router function $\mathcal{S}(x)$:

$$f_{\text{SMoE}}(\boldsymbol{x}) = \sum_{i=1}^{N} \mathcal{S}(\boldsymbol{x})_i \cdot E_i(x) = \sum_{i=1}^{N} \mathcal{S}(\boldsymbol{x})_i \cdot \boldsymbol{W}_{\text{FFN}_i}^2 \phi\left(\boldsymbol{W}_{\text{FFN}_i}^1 \boldsymbol{x}\right) \tag{1}$$

Where $\mathcal{S}(x)$ is computed by $TopK$ function as equation (2) that determines the contribution of each expert to the SMoE output.

$$\mathcal{S}(\boldsymbol{x}) = \text{TopK}(\text{softmax}(\mathcal{G}(\boldsymbol{x})), k); \text{TopK}(\boldsymbol{v}, k) = \begin{cases} \boldsymbol{v_i} & \text{if } \boldsymbol{v_i} \text{ is in the top } k \text{ largest of } \boldsymbol{v} \\ -\infty & \text{otherwise} \end{cases} \tag{2}$$

**Discrete Representation Learning.** van den Oord et al. (2017) propose VQVAE, which uses Vector Quantisation (VQ) to learn a discrete representation. Given an input $x \in \mathbb{R}^{n \times d}$, VQVAE discretized the input into a codebook $V \in \mathbb{R}^{K \times d}$ where $K$ is the codebook size and $d$ is the dimension of the embedding. Let denote $z_v(x) \in \mathbb{R}^{n \times d}$ denotes the output of the VQVAE and $\mathbf{1}()$ is the indicator function. The discrete representation $z_q(x_i) = v_k$, where $k = \arg\min_j \|z_v(x_i) - v_j\|_2$ is achieved by vector quantizer $q_\theta$ that maps an integer $z$ for each input $x$ as:

$$q_\theta(z = k \mid x) = \mathbf{1}\left(k = \underset{j=1:K}{\arg\min} \|z_v(x) - \text{V}_j\|_2\right) \tag{3}$$

### 3.2 VECTOR-QUANTIZED MIXTURE OF EXPERTS (VQMoE)

**Pre-training VQMoE.** Existing Sparse Mixture of Experts (SMoE) models learn continuous representations and select experts based on routing scores derived from token-expert embeddings. In this paper, we propose a novel architecture that learns simultaneously continuous and discrete representations at a training phase as Figure 1a. The continuous representation enables the model to capture complex structures in the data, while the discrete representation learns latent representation from data and then transfers the knowledge to downstream tasks. Given $\mathbf{x} \in \mathbb{R}^{n \times d}$ as the output of the MHA and $f^{\text{v}}$ is a vector quantization operator, the output of the VQMOE layer at the Pre-training phase as follows:

$$f^{\text{VQMoE}}(\boldsymbol{x}) = g(x)_c f^{\text{SMoE}}(\boldsymbol{x}) + g(x)_d \sum_{l=1}^{K} f_l^{\text{FFN}}(\tilde{\boldsymbol{x}_l}), \tag{4}$$

Where $\tilde{x}_l = v_k$ if $x_l \in V_l$ codebook, otherwise $\tilde{x}_l = \vec{0}$ ; $f_l^{\text{FFN}}(\tilde{x}_l)$ corresponds to the expert associated with the $V_l$ codebook; $g(x)_c(x) = col_0(G(x))$, $g(x)_d(x) = col_1(G(x))$ is gating function for continuous and discrete representation with $G(x) = \text{softmax}(W_g^T \times x)$. $W_g^T \in \mathbb{R}^{2 \times d}$ is a learnable weight and $K$ is number of codes.

**Fine-tuning VQMoE.** According to (Geva et al., 2021), the Feed-forward layers (FFN) constitute two-thirds of a transformer model's parameters. Thus, VQMoE enhances the robustness and efficiency of the Mixture of Experts by leveraging the discrete representations learned during the Pre-training phase. For further details, the output of VQMoE during the fine-tuning stages only requires the discrete representation part as Figure 1b, leading to the following output from the VQMoE layer in the fine-tuning phase:

$$f^{\text{VQMoE}}(\boldsymbol{x}) = \sum_{l=1}^{K} f_l^{\text{FFN}}(\tilde{\boldsymbol{x}_l}) \tag{5}$$

### 3.3 TRAINING PROCEDURE

**Pretraining.** The training objective is jointly minimizing the loss of the target task and losses of the Vector Quantization module ($\mathcal{L}^{l2}$ and $\mathcal{L}^{\text{commitment}}$) as in (van den Oord et al., 2017). Equation

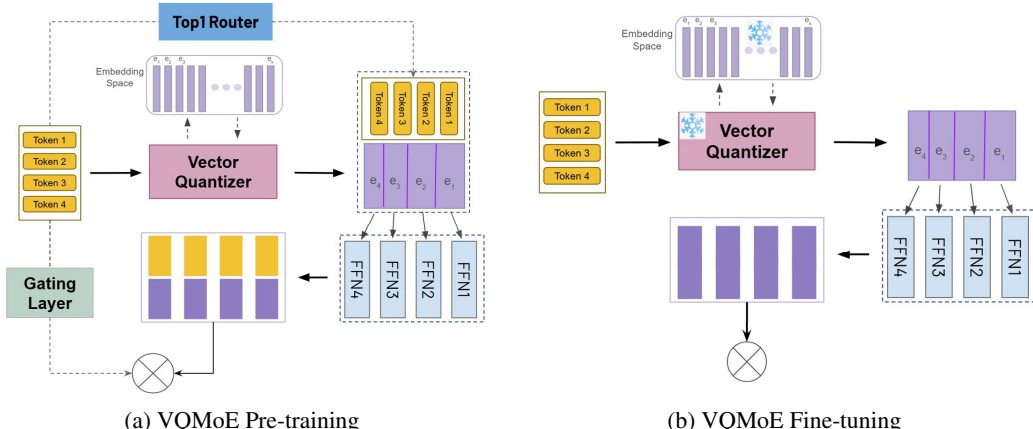

(a) VQMoE Pre-training

(b) VQMoE Fine-tuning

Figure 1: Illustration of the proposed VQMoE architecture for Pre-training and fine-tuning. (a) At the Pre-training stage, VQMoE architecture learns simultaneously continuous and discrete representation at the Pre-training phase. The continuous representation is learned by the conventional SMoE, while the Vector Quantization block facilitates the learning of a discrete representation. The final output is then combined by a gate layer. (b) VQMoE learns a discrete representation that is capable of operating efficiently and robustly on downstream tasks. VQMoE computes the discrete representation only during the fine-tuning stage to achieve robustness and efficiency.

6 specifies the overall loss function for training VQMoE with three components: (1) task loss; (2) $l_2$ loss; (3) a commitment loss. While $\mathcal{L}^{l2}$ helps to move the embedding $v_i$ towards the outputs $z_v(x)$, the commitment loss makes sure the output of the Vector Quantization module commits to the embedding and its output does not grow. The Vector Quantization algorithm does not vary with $\beta$, we follow $\beta = 0.25$ as van den Oord et al. (2017). We introduce a new parameter, $\alpha$, to regulate the contribution of the Vector Quantization loss to the overall loss. A higher value of $\alpha$ favors a stronger adherence to the discrete representation, and vice versa.

$$L = \mathcal{L}_{\text{task}} + \alpha(\|\text{sg}\left[z_v(x)\right] - v\|_2^2 + \beta \|z_v(x) - \text{sg}[v]\|_2^2) \tag{6}$$

where $sg(.)$ is the stop gradient operator defined as follows:

$$\text{sg}(x) = \begin{cases} x & \text{forward pass} \\ 0 & \text{backward pass} \end{cases} \tag{7}$$

**Fine-tuning.** For downstream tasks, we fine-tune the pretraining model by utilizing the codebook learned from the Equation 6 by freezing all parameters at the Vector Quantization module. Thus, the training objective simply becomes: $L = \mathcal{L}_{\text{task}}$.

## 4  THEORETICAL GUARANTEES OF VQMoE

### 4.1  THEORY ANALYSIS

**Problem settings.** We consider an MoE layer with each expert being an MLP layer which is trained by gradient descent and input data $\{(\mathbf{x}_i, y_i)\}_{i=1}^{n}$ generated from a data distribution $\mathcal{D}$. Same as (Chen et al., 2022); (Dikkala et al., 2023), we assume that the MoE input exhibits cluster properties, meaning the data is generated from $K$ distinct clusters $(C_1, C_2, ..., C_k)$.

Inspired by (Dikkala et al., 2023), we conceptualize the router in Sparse Mixture of Experts as a clustering problem. This leads us to define a consistent router in Definition B.1. Furthermore, we introduce a definition for an inconsistent router in SMoE as outlined in Definition B.2, along with the concept of inconsistent expert selection presented in Theorem 4.1 during the training of SMoE.

**Theorem 4.1 (Inconsistent Experts Selection)** *Let $f_{MHA}$ be a multi-head attention (MHA) function producing an output $x \in \mathbb{R}^{n \times d}$, and consider $N$ experts with embeddings $e_i$ for expert $i$ where*

$i \in [1, N]$. *Assume that $f_{MHA}$ converges at step $t_m$, while the expert embeddings $e$ converge at step $t_e$, with $t_m \gg t_e$. For each output $x$, the expert $K \in [1, N]$ is selected such that*

$$K = \arg \min_{j \in [1, N]} dist(x, e_j).$$

*Under these conditions, the expert embeddings $e$ form an inconsistent routing mechanism.*

The proof of Theorem 4.1 is given in Appendix A, and we have the following insights. Theorem 4.1 implies that an expert selection process by a router as the conventional SMoE leads to the inconsistent router. Indeed, the router layer is designed as a simple linear layer, $x$ is the output of MHA function in practice. In practice, an SMoE router is significantly simpler than the MHA function. Consequently, this design leads to the router functioning as an inconsistent router, contributing to the representation collapse issue and instability during training.

**Proposition 4.2 (Optimal Experts Selection)** *Given input data partitioned into $k$ clusters $(C_1, C_2, \ldots, C_k)$ and a mixture of experts (MoE) layer with $k$ experts $(E_1, E_2, \ldots, E_k)$, the assignment of each cluster $C_i$ to expert $E_i$ for $i \in [1, k]$ constitutes an optimal expert selection solution.*

Proposition 4.2 demonstrates that if we are given a clustering structure as input, assigning each part of the input to its corresponding expert results in an optimal expert selection. This implies that learning a discrete representation and directing each component to the appropriate expert yields an optimal solution. The proof of Proposition 4.2 can be found in Appendix A.

## 4.2 VQMoE solves Representation Collapse by Design

The representation collapse problems in SMoE, which leads all experts to learn the same thing, first declared by (Chi et al., 2022). Same as (Chi et al., 2022); (Do et al., 2023), we illustrate the presentation collapse issue by Jacobian matrix approach. Indeed, Jacobian matrix of SMoE with respect to $x \in \mathbb{R}^{n \times d}$ is followed as:

$$\boldsymbol{J_{SMoE}} = \mathcal{S}(x)_k \boldsymbol{J}^{\text{FFN}} + \sum_{j=1}^{N} \mathcal{S}(x)_k \left(\delta_{kj} - S_j\right) \boldsymbol{E}(x)_i \boldsymbol{e}_j^\top = \mathcal{S}(x)_k \boldsymbol{J}^{\text{FFN}} + \sum_{j=1}^{N} \boldsymbol{c}_j \boldsymbol{e}_j^\top, \quad (8)$$

where $\boldsymbol{c}_j = \mathcal{S}(x)_k \left(\delta_{kj} - S_j\right) \boldsymbol{E}(x)_i$. Equation 8 consists two terms: (1) $\mathcal{S}(x)_k \boldsymbol{J}^{\text{FFN}}$ represents a contribution from input token and experts to the final output; (2) $\sum_{j=1}^{N} \boldsymbol{c}_j \boldsymbol{e}_j^\top$ indicates to learn better gating function to minimize the task loss. Moreover, Equation 8 is suggested to be updated toward a linear combination of the expert embeddings. Since $N << d$ in practice, the above equation shows representation collapse from $\mathbb{R}^d$ to $\mathbb{R}^N$.

Compared to SMoE, does VQMoE reduce the representation collapse issue? To answer the essential question, we calculate the Jacobian matrix of VQMoE with respect to $x \in \mathbb{R}^{n \times d}$ is given by:

$$\boldsymbol{J_{VQMoE}} = g\left(x\right)_c J_{SMoE} + J_{g(x)_c} f_{\text{SMoE}}(x) + g\left(x\right)_d J_{VQ} + J_{g(x)_d} f_{\text{VQMoE}}(x) \quad (9)$$

Equation 9 is written shortly as below:

$$\boldsymbol{J_{VQMoE}} = J_1 + \sum_{j=1}^{N} c_j e_j^\top + \sum_{l=1}^{K} d_l e_l^\top + \sum_{m \in c, d} g_m e_m^\top = J_1 + \sum_{j=1}^{N+K+2} o_j e_j^\top \quad (10)$$

where $J_1 = \mathcal{S}(x)_k \boldsymbol{J}^{\text{FFN}}$ ; $c_j = \mathcal{S}(x)_k \left(\delta_{kj} - S_j\right) \boldsymbol{E}(x)_i$ ; $d_l = g\left(x\right)_d$ (due to the vector quantization operator using pass gradient trick (van den Oord et al., 2017)); $g_m = \mathcal{S}(x)_m \left(\delta_{mk} - S_k\right) f_m$ *where* $f_m \in [f_{\text{SMoE}}(x), f_{\text{VQMoE}}]$.

Same as the Jacobian matrix of SMoE, the Jacobian matrix of VQMoE consists two terms: (1) $J_1$ depends on input token and experts to the final output; (2) $\sum_{j=1}^{N+K+2} o_j \boldsymbol{e}_j^\top$ indicates to learn better gating function to minimize the task loss. We can see that $N + K + 2 >> N$, it implies that VQMoE is better than SMoE to solve the representation collapse issue. In theory, we can choose the number of codes to be approximately $d - N - 2$ with a hashing index to experts to address the issue. However, this involves a trade-off with the computational resources required to learn the codebook.

| Configuration | | Enwik8 (BPC) | | Text8 (BPC) | | WikiText-103 (PPL) | | lm1b (PPL) | |
|---|---|---|---|---|---|---|---|---|---|
| Architecture | Algorithm | Base | Large | Base | Large | Base | Large | Base | Large |
| Transformer | VQMoE | **1.48** | **1.41** | **1.47** | **1.40** | **38.74** | **31.98** | **59.48** | **49.30** |
| | SMoE | 1.49 | 1.41 | 1.49 | 1.40 | 39.50 | 32.30 | 60.88 | 51.30 |
| | SMoE-Dropout | 1.82 | 2.22 | 1.70 | 1.89 | 72.62 | 107.18 | 97.45 | 159.09 |
| | XMoE | 1.51 | 1.42 | 1.49 | 1.42 | 39.56 | 32.65 | 61.17 | 51.84 |
| | StableMoE | 1.49 | 1.42 | 1.49 | 1.41 | 39.45 | 32.34 | 60.72 | 50.74 |
| Transformer-XL | VQMoE | **1.19** | **1.08** | **1.28** | **1.17** | **29.48** | **23.85** | **56.85** | **48.70** |
| | SMoE | 1.20 | 1.09 | 1.29 | 1.18 | 30.16 | 24.02 | 58.00 | 48.71 |
| | SMoE-Dropout | 1.56 | 2.24 | 1.56 | 1.86 | 58.37 | 40.02 | 93.17 | 68.65 |
| | XMoE | 1.21 | 1.09 | 1.28 | 1.17 | 30.34 | 24.22 | 58.33 | 50.64 |
| | StableMoE | 1.20 | 1.10 | 1.28 | 1.19 | 29.97 | 24.19 | 58.25 | 49.17 |
| # Params | | 20M | 210M | 20M | 210M | 20M | 210M | 20M | 210M |

Table 1: BPC on the enwik-8 and text8 test sets; and perplexity on the Wikitext-103 and One Billion Word test sets. Lower is better, best results are in bold.

## 5 EXPERIMENT

We conduct experiments to explore the following hypotheses: (i) VQMoE provides an effective SMoE training algorithm for LLMs; (ii) VQMoE delivers a robust and efficient solution during the fine-tuning phase; and (iii) VQMoE outperforms other routing methods in vision domain.

### 5.1 EXPERIMENTAL SETTINGS

To answer the three above hypotheses, we conduct experiments on Vision, Language, and Time-series tasks. For **Pre-training language models**, we examine two common tasks in the training and evaluation of large language models: character-level language modeling using the enwik8 and text8 datasets (Mahoney, 2011), and word-level language modeling with the WikiText-103 (Merity et al., 2016) and One Billion Word datasets (Chelba et al., 2014). For **Parameter-efficient fine-tuning**, we consider pre-trained base models on enwik8 and efficient Fine-tuning it on a downstream task. We choose the SST-2 (Socher et al., 2013), SST-5 (Socher et al., 2013), IMDB (Maas et al., 2011), and BANKING77 (Casanueva et al., 2020) datasets. For **vision tasks**, we employ the Vision Transformer model (Dosovitskiy et al., 2021) with the state-of-the-art routing method and our method to train and evaluate the image classification task. Our experiments encompass four image classification datasets: Cifar10, Cifar100 (Krizhevsky, 2009), STL-10 (Coates et al., 2011), SVHN (Netzer et al., 2011).

### 5.2 PRE-TRAINING LANGUAGE MODELS

**Training tasks** We explore two common tasks in the training and evaluation of LLMs. First, character-level language modeling on the enwik8 or text8 datasets (Mahoney, 2011), which are common datasets to evaluate the model's pre-training capabilities. We also consider the word-level language modeling task on WikiText-103 (Merity et al., 2016) and One Billion Word dataset (Chelba et al., 2014), a much larger and more challenging dataset, to test the models scaling capabilities. For all datasets, we follow the default splits of training-validation-testing. Second, we consider Fine-tuning the models on downstream applications to investigate the models' capabilities of adapting to different domains. To this end, we consider pre-trained medium models on enwik8 and Fine-tuning them on a downstream task. We choose the SST-2 (Socher et al., 2013), SST-5 (Socher et al., 2013), IMDB (Maas et al., 2011), and BANKING77 (Casanueva et al., 2020) datasets, which are common NLP tasks to evaluate pre-trained models. Following Chen et al. (2023a), we freeze the router and only optimize the experts' parameter in this experiment.

**Models.** For the language tasks, we follow the same settings as in SMoE-Dropout (Chen et al., 2023a). We consider two decoder-only architectures: (i) the standard Transformer (Vaswani et al., 2017); and (ii) and Transformer-XL (Dai et al., 2019a) with the same number of parameters as Transformer. We evaluate our method versus the state of art Sparse Mixture of Expert Layers such as StableMoE (Dai et al., 2022) and XMoE (Chi et al., 2022). We consider two model configurations: (i) base: with

four SMoE blocks and **20M** parameters; (ii) large: with twelve SMoE layers and **210M** parameters. We emphasize that we are not trying to achieve state-of-the-art results due to the limited resource constraints. Instead, we evaluate the small and large models on various datasets to demonstrate the scalability and efficacy of our algorithm. Lastly, we conduct extensive investigations using the tiny model to understand the algorithm behaviours and their robustness to different design choices. Lastly, unless otherwise stated, we implement them with $K = 2$ in the experiments.

**Baselines.** We compare our VQMoE with state-of-the-art SMoE training strategies for LLMs. **SMoE** (Jiang et al., 2024) employs a simple router trained end-to-end with the experts. **Stable-MoE** (Dai et al., 2022) proposes a two-phase training process where the first phase trains only the router, and then the router is fixed to train the experts in the second phase. **XMoE** (Chi et al., 2022) implements a deep router that comprises a down-projection and normalization layer and a gating network with learnable temperatures. Lastly, motivated by SMoE-Dropout (Chen et al., 2023a), we implement the **SMoE-Dropout** strategy that employs a randomly initialized router and freeze it throughout the training process.

**Training procedure.** For the language modeling experiments, we optimize the base models and the large models for 100,000 steps. We use an Adam (Kingma & Ba, 2017) optimizer with a Cosine Annealing learning rate schedule (Loshchilov & Hutter, 2017). The lowest validation loss checkpoint is used to report the final performance on the test set.

***Q1: Does VQMoE perform better on Pre-training tasks compared to routing methods? A1: Yes.***

Table 1 presents the evaluation metrics comparing VQMoE with state-of-the-art approaches. We also show the performance progression of the base model on the validation set. Notably, across all methods, the Transformer-XL architecture consistently outperforms the standard Transformer on all datasets. While advanced strategies like XMoE and StableMoE tend to surpass vanilla SMoE when model complexity is increased (from small to medium) or more data is introduced (moving from enwik8 to WikiText-103 or One Billion Word), these improvements are often inconsistent or marginal. In contrast, VQMoE consistently outperforms all competitors across benchmarks (keeping in mind that the BPC metric is log-scaled), architectures, and also converges more quickly. This highlights VQMoE's effectiveness in learning an efficient routing policy for the language modeling pre-training task.

***Q2: Does VQMoE keep outperforming the router method when scaling up? A2: Yes.***

Table 1 also demonstrates that VQMoE maintains consistently strong performance when scaled up to 12-layer Transformer and Transformer-XL architectures. Across all four datasets, the performance gap between VQMoE and other routing methods widens as the dataset size increases, from enwik8 to the One Billion Word dataset. This suggests that our approach has the potential to scale effectively with larger language models and bigger datasets. An interesting observation is that SMoE-Dropout (Chen et al., 2023a) performs the worst among all methods, indicating that a random routing policy is insufficient and requires updating for effective training. This finding highlights that the success of SMoE-Dropout is largely due to its self-slimmable strategy, which linearly increases the number of activated experts ($K$) during training. However, this approach transforms the sparse network into a dense one, contradicting the original motivation behind using SMoE for large-scale models.

***Q3: When does VQMoE outperform router methods in terms of robustness? A3: The lower hidden size of FFN.***

Compared to the routing methods, VQMoE achieves competitive performance which only requires 80% number of parameters. Figure 2a and Figure 2b demonstrate the robustness of our method on the Enwik8 and Text8 datasets, respectively.

## 5.3 PARAMETER-EFFICIENT FINE-TUNING

***Q4: What is the biggest advantage of SMoE, compared to the conventional SMoE? A4: Parameter-Efficient Fine-Tuning.***

We see that the discrete representation that VQMoE learns at the Pretraning stage 5.2 might consist of rich knowledge. To test this hypothesis, we use only the discrete representation for downstream tasks, allowing VQMoE to **save 28%** of computational resources compared to SMoE. Table 2 reports the accuracy of the models fine-tuned on the test sets of various datasets. Overall, we observe that

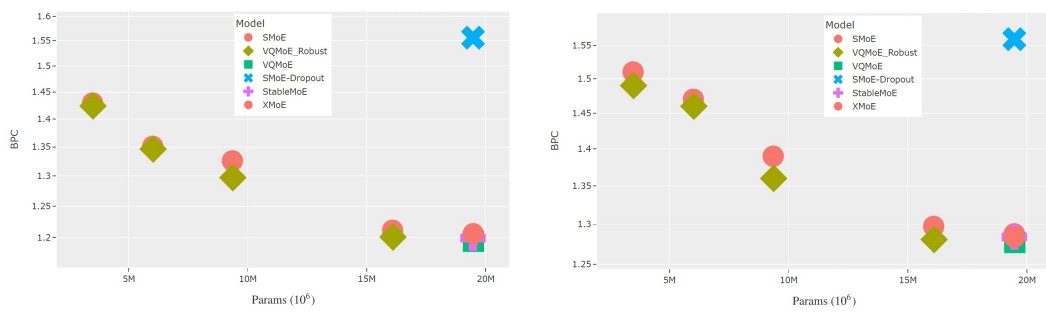

(a) Robust VQMoE Benchmark (Enwik8)      (b) Robust VQMoE Benchmark (Text8)

Figure 2: Illustration of the proposed Robust VQMoE architecture for Pre-training on Enwik8 and Text8 dataset. (a) Robust VQMoE architecture achieves the same performance with the routing methods while only using 80% of the parameters on Enwik8 dataset. (b) Roubust VQMoE demonstrates robustness on the Text8 dataset. Bits-per-character (BPC) on the Enwik8 and Text8 datasets, and lower is better.

| Architecture | FLOPs(x$10^{10}$) | Transformer | | | | Transformer-XL | | | |
|---|---|---|---|---|---|---|---|---|---|
| Dataset | | SST-2 | SST-5 | IMDB | BANKING77 | SST-2 | SST-5 | IMDB | BANKING77 |
| VQMoE | **5.6145** | **82.6** | **41.1** | **89.5** | **84.8** | **83.3** | **42.0** | **89.1** | **85.3** |
| SMoE | 7.7620 | 82.1 | 39.5 | 89.3 | 82.6 | 80.8 | 40.4 | 88.6 | 80.2 |
| SMoE-Dropout | 7.7620 | 81.3 | 39.6 | 88.9 | 77.9 | 81.8 | 40.0 | 89.1 | 77.3 |
| XMoE | 7.7620 | 82.4 | 39.9 | 89.0 | 83.1 | 81.3 | 40.3 | 88.7 | 82.7 |
| StableMoE | 7.7620 | 82.2 | 40.4 | 89.1 | 82.7 | 82.5 | 41.1 | 88.5 | 78.6 |

Table 2: Accuracy of the model after fine-tuned on various datasets. Higher is better, best results are in bold.

VQMoE demonstrates strong transfer learning capabilities by achieving the highest accuracy on all datasets. Notably, on the more challenging datasets of SST-5 and BANKING77, which have fewer training samples or more classes, we observe larger performance gains from VQMoE versus the remaining baselines (over $5\%$ improvements compared to the second-best method). This result shows that VQMoE can learn a discrete representation that is not only good for pre-training but also exhibits strong transfer capabilities to various downstream tasks.

## 5.4 VISION

### Q5: Can VQMoE compete with SMoE in the Vision domain? A5: Yes.

To make our performance comparison informative and comprehensive, we consider two kinds of baselines that are fairly comparable to VQMoE: (1) Dense Model (Vision Transformer) (Dosovitskiy et al., 2021); (2) SoftMoE (Puigcerver et al., 2024) - the most advanced MoE in Vision domain. We perform two configurations for training the Mixture of Experts: (1) small - *10 million parameters (10M)*; (2) *large - 110 million parameters (110M)*. The result at Table 3 shows that VQMoE outperforms both Vision Transformer Dense (Dosovitskiy et al., 2021), SoftMoE (Puigcerver et al., 2024), , and other routing methods such as (Dai et al., 2022), (Chi et al., 2022) on six out of eight tasks across four image classification datasets. We also run our experiments three times with different seeds and report the average result and standard deviation. The average performance of our method surpasses other baselines and is more stable, as indicated by the low standard deviation.

## 5.5 IN-DEPTH ANALYSIS

**Consistent Score.** Figure 3a illustrates that expert selections when training SMoE face inconsistent problems. As the Theorem 4.1, this inconsistency arises because the router's coverage rate significantly exceeds that of the Transformer representation. The figure 3a also shows that our method achieves the highest consistency score compared to the SMoE and XMoE models. However, the

| Architecture | Vision Transformer (Small) | | | | Vision Transformer (Large) | | | | Average |
| # params | 10M | | | | 110M | | | | - |
| Dataset | Cifar10 | Cifar100 | STL-10 | SVHN | Cifar10 | Cifar100 | STL-10 | SVHN | - |
| VQMoE | **89.7**$_{\pm0.4}$ | **67.3**$_{\pm0.4}$ | 66.5$_{\pm0.3}$ | **95.6**$_{\pm0.1}$ | **92.8**$_{\pm0.3}$ | **67.0**$_{\pm0.5}$ | 64.3$_{\pm0.5}$ | **96.0**$_{\pm0.2}$ | **79.9**$_{\pm0.3}$ |
| SMoE | 88.7$_{\pm0.2}$ | 65.4$_{\pm0.5}$ | 66.4$_{\pm0.1}$ | 95.4$_{\pm0.1}$ | 85.7$_{\pm8.5}$ | 55.5$_{\pm2.8}$ | 64.4$_{\pm0.2}$ | 94.5$_{\pm0.1}$ | 77.0$_{\pm1.6}$ |
| XMoE | 88.8$_{\pm0.2}$ | 65.5$_{\pm0.5}$ | 66.3$_{\pm0.2}$ | 95.4$_{\pm0.1}$ | 87.1$_{\pm6.4}$ | 55.9$_{\pm0.6}$ | **64.6**$_{\pm0.3}$ | 94.1$_{\pm0.2}$ | 77.2$_{\pm1.1}$ |
| StableMoE | 88.8$_{\pm0.1}$ | 65.5$_{\pm0.1}$ | 66.5$_{\pm0.2}$ | 95.4$_{\pm0.1}$ | 84.7$_{\pm10.5}$ | 55.5$_{\pm1.8}$ | 64.3$_{\pm0.6}$ | 94.5$_{\pm0.9}$ | 76.9$_{\pm1.8}$ |
| SoftMoE | 85.6$_{\pm0.3}$ | 61.4$_{\pm0.3}$ | 65.4$_{\pm0.2}$ | 94.8$_{\pm0.1}$ | 80.3$_{\pm9.7}$ | 42.9$_{\pm1.4}$ | 63.2$_{\pm0.5}$ | 93.5$_{\pm0.1}$ | 73.4$_{\pm1.6}$ |
| ViT (Dense) | 89.0$_{\pm0.2}$ | 65.7$_{\pm0.3}$ | **66.6**$_{\pm0.2}$ | 95.6$_{\pm0.1}$ | 92.2$_{\pm0.3}$ | 60.2$_{\pm2.6}$ | 64.1$_{\pm0.5}$ | 96.0$_{\pm0.1}$ | 78.7$_{\pm0.5}$ |

Table 3: Accuracy of models evaluated on vision datasets. Higher is better, best results are in bold.

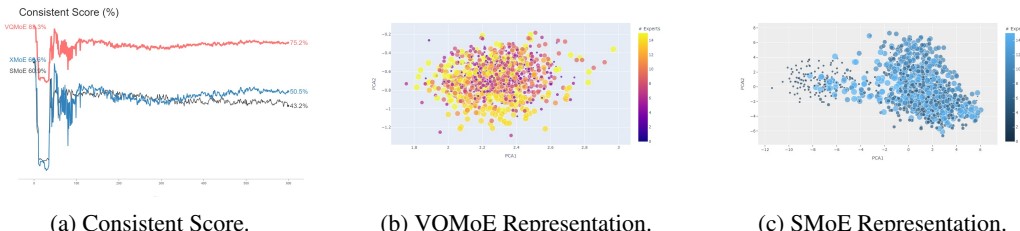

(a) Consistent Score.    (b) VQMoE Representation.    (c) SMoE Representation.

Figure 3: Analysis Inconsistent Expert Selection and Representation Collapse issues when training SMoE. Figure 3a demonstrates consistent score movement from VQMoE, compared with SMoE and XMoE. Figure 3b and Figure 3c visualize the representation by experts in 2D dimension using Principal Component Analysis (PCA) method.

VQMoE model's consistency score is around 75%, as our method also requires learning a continuous representation during the Pre-training phase.

**Representation Collapse issue.** To visualize the Representation collapse problem in practice, we apply Principal Component Analysis (PCA) method to reduce from $d$ dimension of the Transformer to 2D for plotting purposes, thanks to (Chi et al., 2022). Figures 3b and 3c show the expert representations from the pretrained VQMoE and SMoE models. The results suggest that VQMoE experiences less representation collapse in the expert space compared to SMoE. The analysis is in line with the theorem proof at Section 4.2. However, projecting the $d$-dimensional space onto 2D for visualization may lead to information loss.

## 5.6 ABLATION STUDY

We examine the effectiveness of VQMoE across various hyper-parameter settings, with all experiments conducted using the base Transformer architecture on the WikiText-103 dataset.

**Vector Quantization Method.** To learn a discrete representation, we research various types of Vector Quantization methods, including VQVAE (van den Oord et al., 2017), VQGAN (Yu et al., 2022), LFQ (Yu et al., 2023), and ResidualVQ (Yang et al., 2023). We observe that VQGAN using cosine similarity for distance achieves good and stable results in practice as Figure 6a. Interestingly, VQGAN with lower dimensionality also delivers strong performance and exhibits robustness.

**Number of codebook impact.** The number of codebook entries is a crucial hyperparameter when training Vector Quantization techniques. As shown in Figure 6b, we can see the best performance when the number of codebook entries matches the number of experts. This aligns with the proof by (Dikkala et al., 2023), which demonstrates that in the optimal case, the number of clusters equals the number of experts.

**Sensitiveness of VQ loss contribution** $\alpha$. Figure 6c illustrates the impact of $\alpha$, which controls the contribution of the Vector Quantization loss to the overall loss. If $\alpha$ is too high, it leads to a better discrete representation but may negatively affect the final target. Conversely, if $\alpha$ is too low, it may result in a poor discrete representation. Therefore, $\alpha$ should be selected based on the data, typically within the range of $(0.05, 0.15)$.

# 6 CONCLUSION AND FUTURE DIRECTIONS

This study illustrates Vector-Quantized Mixture of Experts (VQMoE), which is novel and theoretically-grounded architecture to overcome challenges in training SMoE such as representation collapse and inconsistency. We evaluate our method on various Pre-training and Fine-tuning tasks, for both language and vision domains. The results show that VQMoE outperforms the routing methods both theoretically and empirically. Furthermore, fine-tuning VQMoE with the discrete representation for downstream tasks could reduce computational resource usage by 28%. We believe that focusing on discrete representation learning will offer a promising strategy for training and testing sparse mixtures of experts (SMoE) at a large scale. Finally, we believe that our approach opens up new research avenues for effectively training SMoE, where cutting-edge techniques in discrete representation learning and vector quantization can be harnessed to enhance their performance.

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

# A    APPENDIX

# Supplementary Material for "On the effectiveness of discrete representations in sparse mixture of experts"

This document is organized as follow. Appendix B presents the detailed proof of our theoretical analysis in Section 4. Appendix C provide in-depth analysis about the representation collapse while Appendix D presents all the implementation details and additional results.

# B    PROOF FOR RESULTS IN SECTION 4

## B.1    PROOF OF THEOREM 4.1

**Definition B.1 (Consistent Router)** *A sequence of points $x_1, x_2, \ldots, x_n$ and a corresponding sequence of clusters $C_1, C_2, \ldots, C_k$ are said to be **consistent** if, for every point $x_p \in C_i$, the condition*

$$dist(x_p, u_i) \leq \min_{j \neq i} dist(x_p, u_j)$$

*is satisfied, where $dist(a, b)$ denotes the distance between $a$ and $b$, and $u_i$ is the center of cluster $C_i$.*

**Definition B.2 (Inconsistent Router)** *A sequence of points $x_1, x_2, \ldots, x_n$ and a corresponding sequence of clusters $C_1, C_2, \ldots, C_k$ are said to be **inconsistent** if there exists a point $x_p \in C_i$ such that*

$$dist(x_p, u_i) > \min_{j \neq i} dist(x_p, u_j),$$

*where $dist(a, b)$ represents the distance between $a$ and $b$, and $u_i$ is the center of cluster $C_i$.*

In this proof, we use contradiction to establish the theorem. Assume that the expert embeddings $e$ form a consistent router. By Definition B.1, we have:

$$\mathrm{dist}(x_p, u_i) \leq \min(\mathrm{dist}(x_p, C_j)),$$

where $u_i$ is the representation corresponding to the closest expert $e_i$.

According to (Chi et al., 2022), projecting information from a hidden representation space $\mathcal{R}^d$ to the expert dimension $N$ leads to representation collapse. Now, consider three experts $x, y, z$ whose embeddings $e_x, e_y, e_z$ collapse. Without loss of generality, assume that $e_y$ lies between $e_x$ and $e_z$ in the embedding space. Then, we have:

$$\mathrm{dist}(y, u_y) \leq \min(\mathrm{dist}(x, e_x), \mathrm{dist}(y, e_y), \mathrm{dist}(z, e_z)) \leq \mathrm{dist}(e_x, e_z).$$

Let $t_e$ denote the step at which the embeddings $e_x$ and $e_z$ converge, and $t_m$ denote the step at which the Multi-Head Attention (MHA) module converges. From step $t_e$, it follows that:

$$\lim_{t_e \to t_m} \mathrm{dist}(y, u_y) = \lim_{t_e \to t_m} \mathrm{dist}(e_x, e_z) = 0.$$

Thus, $y$ (the output of MHA) converges at step $t_e$.

This directly contradicts the assumption that the MHA converges at step $t_m$, where $t_e \ll t_m$.

## B.2 Proof of Proposition 4.2

We use contradiction to prove the proposition. Assume that, at training step $t$, there exists a set of pairs $(C_i, E_j)$ such that $i \neq j$. Let $x_1, x_2, \ldots, x_k$ represent a sequence of inputs sampled from $K$ clusters. From step $t_0$ to step $t_{k-1}$, each pair $(x_j, E_j)$, where $j \in [1, k]$, is updated using the following gradient descent equation:

$$W_{E_j}^{l+1} = W_{E_j}^l - \eta \mathcal{J}(x_j),$$

where $W_{E_j}^l$ is the weight of expert $E_j$ at iteration $l$, $\mathcal{J}(x_j)$ is the Jacobian matrix with respect to input $x_j$, and $\eta$ is the learning rate.

Let $\mathcal{L}$ denote the loss function during the training process described by Equation 6. After $t_k$ training steps, the following condition holds:

$$E_j(x_j) = \min_{c \in [1,k]} E_j(x_c).$$

Under the assumption of contradiction, there exists a set of pairs

$$\sum_{i,j=1; i \neq j}^{K} (C_i, E_j)$$

where the loss function $\mathcal{L}$ is minimized. However, by definition of the loss minimization process, the inequality

$$\sum_{i=1}^{K} (C_i, E_i) \leq \sum_{i,j=1; i \neq j}^{K} (C_i, E_j)$$

must hold.

This leads to a contradiction with our initial assumption.

## C Representation Collapse Analysis

To illustrate Theorem 4.1, we perform a language model task as described in Section D.2, examining the movement of Expert Input Representation in Figure 4 and Expert Embedding (router) in Figure 5. We analyze the dynamics of the expert input representations by tracking their changes across training iterations. The results indicate that the inputs to the experts become increasingly divergent over time. This divergence suggests that the model learns to represent the data in a more specialized and diverse manner, allowing each expert to focus on distinct features or patterns within the data. Similarly, we track the changes in expert embeddings (router) throughout the training process. However, the trend is the opposite: the expert embeddings appear to converge quickly, stabilizing around 10,000 iterations. The findings align with our assumption stated in Theorem 4.1, indicating that Expert Embedding converges more quickly than Expert Input Representation. These results provide further evidence supporting the Theorem 4.1.

## D Experiments implementation details

This section provides detailed parameters of our experiments in Section 5.

### D.1 General Settings

The experiments are based on the publicly available SMoE-Dropout implementation(Chen et al., 2023a)[1]. However, the pre-training was conducted on two H100 GPUs, so results might differ when using parallel training on multiple GPUs.

---

[1] https://github.com/VITA-Group/Random-MoE-as-Dropout

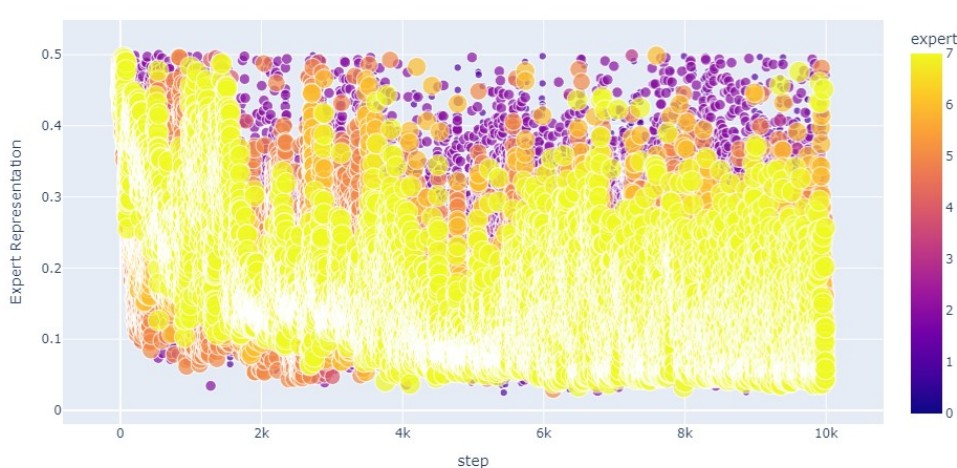

Figure 4: Training SMoE Expert Input Representations across Training Iterations.

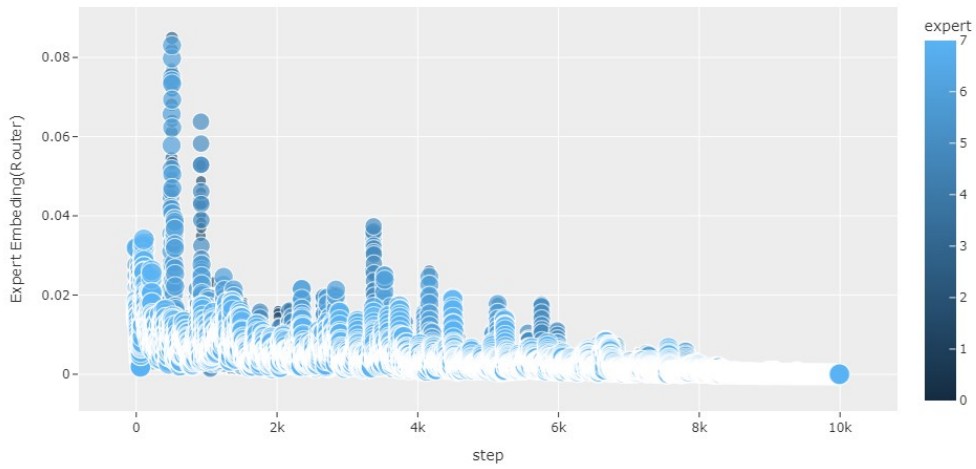

Figure 5: Training SMoE Router (Expert embedding) across Training Iterations.

(a) Vector Quantization method.     (b) Number of codebook.     (c) Impact of $\alpha$ for VQMoE.

Figure 6: Pre-training small Transformer-XL on WikiText-103 across different hyperparameters.

## D.2 PRE-TRAINING EXPERIMENTS

Table 4 provides the detailed configurations for pre-training Transformer (Vaswani et al., 2017), Transformer-XL Dai et al. (2019b) on `Enwik8`, `Text8`, `WikiText-103`, and `One Billion Word`.

| Dataset | Input length | Batch size | Optimizer | Lr | # Training Step |
|---|---|---|---|---|---|
| Enwik8 | 512 | 48 | Adam | 3.5e-4 | 100k |
| Text | 512 | 48 | Adam | 3.5e-4 | 100k |
| WikiText-103 | 512 | 22 | Adam | 3.5e-4 | 100k |
| One Billion Word | 512 | 11 | Adam | 3.5e-4 | 100k |

Table 4: Hyperparameter settings for pre-training experiments on `Enwik8`, `Text8`. , `WikiText-103`. , and `One Billion Word`.

## D.3 FINE-TUNING EXPERIMENTS

For fine-tuning experiments, we employ the identical model architecture as in pre-training. Table 5 presents the detailed configurations utilized for fine-tuning experiments on `SST-2`, `SST-5`, `IMDB`, and `BANKING77` datasets. We start with the pretrained checkpoint of the base model on enwik8, remove the final layer, and replace it with two randomly initialized fully connected layers to serve as the classifier for each fine-tuning dataset. All methods are fine-tuned for 5,000 steps with a uniform learning rate.

| Dataset | Input length | Batch size | Optimizer | Lr | # Epochs |
|---|---|---|---|---|---|
| SST-2 | 512 | 16 | Adam | 1e-4 | 5 |
| SST-5 | 512 | 16 | Adam | 1e-4 | 5 |
| IMDB | 512 | 4 | Adam | 1e-4 | 5 |
| BANKING77 | 512 | 16 | Adam | 1e-4 | 5 |

Table 5: Detail settings for fine-tuning experiments on the evaluation datasets.

