# OpenReview forum: "On the Effectiveness of Discrete Representations in Sparse Mixture of Experts"
_ICLR.cc/2025/Conference — Submitted to ICLR 2025_

### Official Review · Reviewer_WyD6 · 2024-10-30

**Soundness:** 3
**Presentation:** 2
**Contribution:** 3
**Rating:** 5
**Confidence:** 3

**Summary:**

This paper proposes Vector-Quantized Mixture of Experts (VQMoE) as an alternative to SMoE to enhance the efficiency of the Mixture of Expert models. The main objective is to address the "representation collapse" issue, where a small number of experts receive the majority of tokens or all experts converge to learn similar representations, primarily due to the inefficiency of router operations. VQMoE employs a method that assigns experts without a router by using vector quantization. The validity of this approach is theoretically demonstrated, and VQMoE outperforms existing MoE methodologies across various datasets, including both text and visual data.

**Strengths:**

1. The paper propose a novel MoE approach that mitigates dependency on routers, thereby preventing representation collapse and reducing train/inference computation.
2. Theoretical validation of VQMoE's efficiency was attempted (Theorem 4.1, Proposition 4.2).
3. Experiments were conducted on both text and visual datasets, demonstrating that the model performs well even as the model scales up.

**Weaknesses:**

1. There are several weaknesses in the theoretical analysis section:

Theorem 4.1 – The assumption that MHA and expert embeddings converge, converging with t_m>>t_e  does not appear to account for the possibility of representation collapse. A specific mathematical approach addressing the occurrence of representation collapse is missing.

Proposition 4.2 – Here, an even less agreeable assumption is made. It assumes that input data can be divided into k clusters; however, it is well-known in the deep learning field that high-dimensional data, such as text or images, does not cluster cleanly. Furthermore, this claim expects the number of clusters to match the number of experts, which is unlikely to hold for real-world data.

2. In the case of visual experiments, only low-resolution datasets were used, making it difficult to conclude that the performance has been sufficiently validated. Additional experiments on high-res datasets, such as ImageNet, would be necessary to strengthen the findings.

**Questions:**

Please refer to the weaknesses.

---

> ### Author Response · Authors · 2024-11-17
> **Authors response to Reviewer WyD6**
>
> We sincerely appreciate your valuable feedback. Please see our responses to your concerns below.
>
> **W1. Theorem 4.1 – The assumption that MHA and expert embeddings converge, converging with $t_m>>t_e$ does not appear to account for the possibility of representation collapse. A specific mathematical approach addressing the occurrence of representation collapse is missing.**
>
> A1.
>
> As theorem 4.1 The assumption that MHA and expert embeddings converge, converging with $t_m>>t_e$ lead to inconsistent expert selections. From the consistent expert selections to expert representation collapse has been proved by [1].
>
> **W2. Proposition 4.2 – Here, an even less agreeable assumption is made. It assumes that input data can be divided into k clusters; however, it is well-known in the deep learning field that high-dimensional data, such as text or images, does not cluster cleanly. Furthermore, this claim expects the number of clusters to match the number of experts, which is unlikely to hold for real-world data.**
>
> A2.
>
> - We build upon the assumptions of [2] and [3] that the data exhibits a clustering structure. Besides that, other works in MoE, such as [4] and [5], assume that the data contains specific patterns.
> - From a theoretical perspective, [3] demonstrated that the Mixture of Experts (MoE) can be interpreted as a clustering problem. If the number of clusters is smaller than the number of experts, collapse may occur; conversely, if it is larger, the model could underfit. In practice, selecting an appropriate number of experts can be achieved through hyperparameter tuning tailored to the specific dataset.
>
> **W3. Additional experiments on high-res datasets, such as ImageNet, would be necessary to strengthen the findings.**
>
> A3.
>
> We are currently training on the ImageNet dataset and will report the results once the process is complete.
>
> **Reference:**
>
> [1] Chi, Z., Dong, L., Huang, S., Dai, D., Ma, S., Patra, B., Singhal, S., Bajaj, P., Song, X., Mao, X., Huang, H., & Furu Wei, F. (2022) On the representation collapse of sparse mixture of experts https://arxiv.org/abs/2204.09179
>
> [2] Chen, Z., Deng, Y., Wu, Y., Gu, Q., & Li. Y. (2022) Towards understanding mixture of experts in deep learning. https://arxiv.org/abs/2208.02813
>
> [3] Dikkala, N., Ghosh, N., Meka, R., Panigrahy, R., Vyas, N., & Wang. X. (2023) On the benefits of learning to route in mixture-of-experts models. https://aclanthology.org/2023.emnlp-main.583.pdf
>
> [4] Chowdhury, M., Zhang, S., Wang, M., Liu, S., & Chen. P. (2023) Patch-level routing in mixture-of-experts is provably sample-efficient for convolutional neural networks. https://arxiv.org/pdf/2306.04073
>
> [5] Chowdhury, M., Wang, M., Maghraoui, K., Wang, N., Chen, P., &Carothers. C. (2024) A
> provably effective method for pruning experts in fine-tuned sparse mixture-of-experts. https://arxiv.org/abs/2405.16646

---

> > ### Comment · Reviewer_WyD6 · 2024-11-20
> >
> > For A1.
> > I understand that you are trying to make the claim that representation collapse is caused by $t_m>>t_e$. However, even in the paper you referenced [1], there does not seem to be a clear claim for $t_m>>t_e$. You need an experimental or theoretical proof of this.
> > Additionally, I noticed a typo $(j \neq j)$ and missing notation explanation in the proof that follows in Appendix B. This also needs to be corrected.
> >
> > For A2.
> > You have referenced a number of papers that help me understand your work, but this leads me to a more fundamental question: if, as you claim, real data already has a well-clustered structure, and collapse can be solved by tuning the number of experts through hyperparameter tuning, then what is the novelty of your paper? Is it just that existing MoE methodologies suffer from collapse because they fail to properly tune the number of experts? In this respect, I believe that Proposition 4.2 oversimplifies the problem and consequently fails to highlight the advantages of VQMoE as proposed in this paper.
> >
> > For A3.
> > Thank you for your hard work.

---

> ### Author Response · Authors · 2024-11-22
> **[P2] Authors response to Reviewer WyD6**
>
> Dear Reviewer WyD6,
>
> Thank you so much for your question. We would like to answer your questions as below:
>
> **For A1-1. I understand that you are trying to make the claim that representation collapse is caused by $t_m >> t_e$ .You need an experimental or theoretical proof of this.**
>
> A4.
>
> The core theoretical contribution of our work is presented in Section 4.2, “VQMOE Solves Representation Collapse by Design,” where we demonstrate that our method outperforms other routing approaches in addressing representation collapse. Furthermore, we identify inconsistencies in routing selection in practice and support this observation through Theorem 4.1, under the assumption $t_m >> t_e$.  This issue is also illustrated in Figures 4 and 5. Regarding the relationship between representation collapse and inconsistencies in routing, [1] provides visual evidence in Section 4.5 Analysis (Figures 2 and 3).
>
> **For A1-2. Additionally, I noticed a typo and missing notation explanation in the proof that follows.**
>
> A5.
>
> Thank you for your suggestions. We have revised it and updated it in our new revision.
>
> **For A2-1. if, as you claim, real data already has a well-clustered structure,.**
>
> A6.
>
> We think there is a misunderstanding here, our assumption is that data has cluster structure in a latent space, not well-clustered defined structure. Our method will learn the structure by the Vector Quantionzation approach. For example, images or text are unstructured data but they have a cluster structure in latent spaces as illustrated by [1], [2], [3], [4].
>
> ***For A2-2. collapse can be solved by tuning the number of experts through hyperparameter tuning?**
>
> A7.
>
> No, we did not say that, our answer for your first part is "In practice, selecting an appropriate number of experts can be achieved through hyperparameter tuning tailored to the specific dataset."
>
> **For A2-4.  Is it just that existing MoE methodologies suffer from collapse because they fail to properly tune the number of experts?,.**
>
> A8.
>
> No, that is not correct. Even if the true number of experts equals the number of clusters in the hidden space, conventional SMoE still faces the collapse issue due to the projection of information from d_model (dimension of Transformer model) to number_of_experts as proof by [5].
>
> **For A2-5.  What is the novelty of your paper?,.**
>
> A9.
>
> While conventional SMoE models rely on a router to allocate input to each expert, we propose an alternative approach: learning a cluster structure and directing inputs to experts based on this structure, which we find to be more efficient. This is supported by theoretical results in Section 4.2 and experimental results in Section 5.
>
> **For A2-6.  Proposition 4.2 oversimplifies the problem and consequently fails to highlight,.**
>
> A10.
>
> Proposition 4.2 aligns with our method, as it demonstrates that learning a cluster structure and assigning each cluster consistently to the corresponding experts leads to an optimal solution.
>
>
> **W3. Additional experiments on high-res datasets, such as ImageNet, would be necessary to strengthen the findings.**
>
> A11.
>
> VQMoE consistently outperforms both dense models and other routing methods on large-scale datasets like ImageNet, across various scales, from small to large. For further details, please refer to the General Response.
>
> **Reference:**
>
> [1] Li, Y., Hu, P., Peng, D., Lv, J., Fan, J., & Peng. X. (2024) Image clustering with external guidance. https://arxiv.org/abs/2310.11989
>
> [2] Stephan, A., Miklautz, L., Sidak, K., Wahle, J., Gipp, B., Plant, C., & Roth, B. (2024) Text-guided image clustering. https://arxiv.org/abs/2402.02996
>
> [3] Tipirneni, S., Adkathimar, R., Choudhary, N., Hiranandani, G., Amjad, R., N. Ioannidis, V., Yuan, C., & Reddy, C. (2024) Context-aware clustering using large language models. https://arxiv.org/abs/2405.00988
>
> [4] Viswanathan, V., Gashteovski, K., Lawrence, C., Wu, T.,
> & Neubig, G.(2023) Large language models enable few-shot clustering.https://arxiv.org/abs/2307.00524
>
> [5] Chi, Z., Dong, L., Huang, S., Dai, D., Ma, S., Patra, B., Singhal, S., Bajaj, P., Song, X., Mao, X., Huang, H., & Furu Wei, F. (2022) On the representation collapse of sparse mixture of experts https://arxiv.org/abs/2204.09179

---

> > ### Author Response · Authors · 2024-11-24
> >
> > Dear Reviewer WyD6,
> >
> > Thank you for your constructive review. If you have any remaining questions or concerns, please feel free to let us know so we can address them before the deadline. If your concerns have been resolved, we kindly ask that you update your evaluation accordingly.
> >
> > Best,
> >
> > The Authors

---

### Official Review · Reviewer_aPBa · 2024-10-31

**Soundness:** 2
**Presentation:** 2
**Contribution:** 3
**Rating:** 5
**Confidence:** 4

**Summary:**

The article proposes a novel architecture, the Vector-Quantized Mixture of Experts (VQMoE), which aims to address routing inconsistency and representation collapse issues in Sparse Mixture of Experts (SMoE) models, marking the first solution of its kind. Unlike traditional SMoE methods that rely on a router to assign tasks to experts, VQMoE uses discrete representations obtained through vector quantization to select experts. The authors present a theorem demonstrating the existence of routing inconsistency in SMoE and propose a proposition for optimal expert selection. Proofs of the theorem and proposition are provided using counterexamples, and a two-stage training approach is employed. The core idea of VQMoE is to simultaneously learn both continuous and discrete representations, thereby combining the advantages of each. The continuous representation aids the model in capturing complex data details, while the discrete representation quantizes the input, selecting appropriate experts to handle specific tasks. The authors conducted experiments on large language models (LLMs) and vision classification tasks, showing that VQMoE achieves a 28% increase in robustness compared to existing SMoE routing methods.

**Strengths:**

1. This paper is the first to study discrete representations in Sparse Mixture of Experts (SMoE). Unlike traditional SMoE methods that rely on routers to assign tasks to experts, VQMoE uses discrete representations obtained through vector quantization to select experts, addressing routing inconsistency and representation collapse issues in the model.
2. A novel VQMoE architecture is proposed that simultaneously learns both continuous and discrete representations.
3. A theorem and a proposition are presented, with theoretical proofs and experimental validation provided to support them.

**Weaknesses:**

1. In VQVAE, only one codebook is used. Why are multiple codebooks used in VQMoE, and what difference does this make?
2. The traditional SMoE methods have a drawback of routing inconsistency, which the proposed VQMoE aims to address. However, if two tokens are very similar, how does VQMoE ensure they are not quantized to the same codebook?
3. The proofs of the theorem and definitions in the appendix B, particularly Figures 4 and 5, are unclear. It would be helpful if the authors provided a more detailed explanation of these figures.
4. Performance testing on ImageNet is missing from the classification task evaluations.
5. Although the paper states that experiments were conducted in the Vision domain, they are limited to classification tasks, with no experiments in areas like object detection or segmentation.
6. The writing and formula expressions in the paper are sometimes unclear, with some statements potentially leading to misunderstandings. Examples include the description of Equation (3); the subscripts in Equation (1) being in the lower right corner while they appear in the upper right in Equations (4) and (5); the meaning of `where g(x)c + g(x)d` in Equation (4); the reason for two Cifar10 results under Vision Transformer (Large) in Table 3; “Figures 3c and 3c” in Section 5.5; and “where j ≠ j” in Appendix B.1 for the proofs. Additionally, should `min(dist(y, ey))` be represented as `min(dist(y, ex), dist(y, ey), dist(y, ez))`? I hope the authors will carefully revise these issues in the paper and provide more theoretical and experimental support for the key proofs.

**Questions:**

My questions align with the issues raised in the weaknesses section, and I hope the authors will provide thorough explanations.
1. For Weakness 1, please provide an ablation study or theoretical analysis comparing single versus multiple codebooks in VQMoE and explain the rationale behind this design choice.
2. For Weakness 2, please provide experiment results or theoretical analysis of how VQMoE handles very similar tokens.
3. For Weakness 3, please provide step-by-step explanations for Figures 4 and 5, including what each axis represents, how the data points were generated, and how these figures support the theorem and definitions in Appendix B.
4. For Weakness 4, please explain why ImageNet was not included, and please include experiments on ImageNet.
5. For Weakness 5, please include additional experiments on vision tasks such as object detection and image segmentation.
6. For Weakness 6, I hope the authors will carefully review the content to address equation inconsistencies, typographical errors, and unclear explanations.

---

> ### Author Response · Authors · 2024-11-17
> **Authors response to Reviewer aPBa**
>
> Thank you for your thoughtful review! Please find our responses below.
>
> **W1. In VQVAE, only one codebook is used. Why are multiple codebooks used in VQMoE, and what difference does this make?**
>
> A1.
>
> We apologize for the confusion caused by the writing mistake. In our method, We only learn one codebook, $K$ is denoted for the number of codes in the codebook.
>
> **W2.  If two tokens are very similar, how does VQMoE ensure they are not quantized to the same codebook?**
>
> A2.
>
> Proposition 4.2 proves that, for input data with a clustering structure, the optimal expert selection is to route each cluster to its corresponding expert. Thus, if two tokens are very similar, consistently routing them to the same expert is the optimal choice. In contrast, traditional SMoE methods use a router to direct tokens, but as shown in Figure 5, the router converges quickly (typically around 10k iterations), resulting in inconsistent expert selections.
>
> **W3. The proofs of the theorem and definitions in the appendix B, particularly Figures 4 and 5, are unclear. It would be helpful if the authors provided a more detailed explanation of these figures.**
>
> A3.
>
> Thank you for your suggestion and we have updated our revision. Figures 4 and 5 illustrate Theorem 4.1. In Figure 4, we visualize the change in expert representations (FFN) over training iterations, while Figure 5 shows the change in expert embeddings (router) over training iterations. Since the expert embeddings converge faster than the expert representations, this results in inconsistent expert selections.
>
> **W4. Performance testing on ImageNet is missing from the classification task evaluations.**
>
> A4.
>
> We are training the baselines and our method on ImageNet and we will report the result to you as soon as possible.
>
> **W5. Although the paper states that experiments were conducted in the Vision domain, they are limited to classification tasks, with no experiments in areas like object detection or segmentation.**
>
> A5.
>
> Our paper focuses on addressing the representation collapse and inconsistency problems in SMoE foundation models. Applications of the model to tasks such as object detection and segmentation are left for future work. However, [1] demonstrates that a stronger image classification backbone leads to improved performance in both object detection and image segmentation.
>
>
> **W6. The writing and formula expressions in the paper are sometimes unclear, with some statements potentially leading to misunderstandings.**
>
> A6.
>
> Thank you for your valuable suggestions. We have reviewed and revised our paper to enhance its clarity and readability.
>
> **Reference:**
>
> [1] Goldblum, M., Souri, H., Ni, R., Shu, M., Prabhu, V., Somepalli, G., Chattopadhyay, P., Ibrahim, M., Bardes, A., Hoffman, J., Chellappa, R., Wilson, A., & Goldstein, T. (2023) Battle of the backbones: A large-scale comparison of pretrained models across computer vision tasks. https://arxiv.org/abs/2310.19909

---

> > ### Comment · Reviewer_aPBa · 2024-11-20
> > **There are still some questions.**
> >
> > Thanks.
> > 1. For weakness 2, I am concerned about whether there are problems with the quantization mechanism of VQMoE, such as whether highly similar data tokens will lead to the reuse of the same codebook, and the potential loss of information or degradation of generalization ability that may arise from this. **Similar to reviewer WyD6's doubt in weakness1.**
> > In the quantization mechanism of VQMoE, if two data tokens are very similar, will they be forced to be quantized into the same codebook? If yes, will this result in some information loss or model capability degradation? If not, by what specific method does VQMoE differentiate these similar tokens?
> > 2. For weakness 4, I would like to see the effect on ImageNet.
> > 3. For weakness 5, are there relevant experiments on detection or segmentation? Can there be some simple experiments to demonstrate the superiority of your method?
> >
> > Looking forward to your reply.

---

> ### Author Response · Authors · 2024-11-22
> **[P2] Authors response to Reviewer aPBa**
>
> Dear Reviewer aPBa,
>
> Thank you so much for your constructive discussions. We would like to answer your questions as below:
>
> **1. W2-p1 In the quantization mechanism of VQMoE, if two data tokens are very similar, will they be forced to be quantized into the same codebook?**
>
> A7.
>
> Our answer is yes. The Vector Quantization mechanism will learn a codebook to group similar elements by assigning them to the same code within the codebook. Proposition 4.2 proves that if there is an input data with a clustering structure, the optimal expert selection is to route each cluster to its corresponding expert.
>
> **1. W2-p2 will this result in some information loss or model capability degradation?**
>
> A8.
>
> The discrete representation can help a model to learn a true data pattern and improve model generation, however, it can lead to information loss in some cases such as our answer for question 2 of the reviewer ckSP.  To overcome the challenge, we propose to learn both continuous representation and discrete representation for the pretraining model (for large dataset). For downstream tasks (small datasets), we suggest that only discrete representations are efficient in terms of both performance and inference speed.
>
> **2. W4. I would like to see the effect on ImageNet.**
>
> A9.
>
> VQMoE consistently outperforms the dense model and other routing methods on large-scale datasets such as ImageNet, across both small and large-scale versions. Please refer to the General Response for more detail.
>
> **3. W5. Are there relevant experiments on detection or segmentation? Can there be some simple experiments to demonstrate the superiority of your method?**
>
> A10.
>
> For the image segmentation task, our method surpasses the baselines and the dense model in terms of Mean Accuracy and mIoU metrics on the ADE20K dataset using the Segmenter model[1]. Detailed results are provided below.
>
> | Model | ViT | SoftMoe | SMoE | StableMoE | XMoE | VQMoE | Metrics
> | :-------: | :----: | :------:| :--------:| :-------:| :------: | :------:|:-----:|
>  |Segmenter| 66.8 | 65.6 | 66.2 | 67.1 | 66.6 | 66.2| Pixel accuracy|
> |Segmenter| 20.8 | 19.0 | 23.1 | 22.4 | 22.3 | 23.4| Mean accuracy|
> |Segmenter| 15.0 | 14.0 | 15.5 | 16.0 | 15.7 | 16.6| mIoU |
> |Average | 34.2 | 32.9 | 34.9 | 35.2 | 34.9 | 35.4| |
>
> **Reference:**
>
> [1] Robin Strudel, Ricardo Garcia, Ivan Laptev, and Cordelia Schmid. 2021. Segmenter: Transformer for semantic segmentation https://arxiv.org/abs/2105.05633

---

> > ### Author Response · Authors · 2024-11-24
> >
> > Dear Reviewer aPBa,
> >
> > Thank you for your insightful review. If you have any additional questions or concerns, please don’t hesitate to let us know so we can address them before the deadline. If your concerns have been resolved, we kindly request that you update your evaluation accordingly.
> >
> > Best,
> >
> > The Authors

---

### Official Review · Reviewer_YzaE · 2024-11-02

**Soundness:** 3
**Presentation:** 3
**Contribution:** 2
**Rating:** 5
**Confidence:** 4

**Summary:**

This paper introduces Vector-Quantized Mixture of Experts (VQMoE), a novel architecture that addresses the challenges of representation collapse and inconsistency in training Sparse Mixture of Experts (SMoE) by using discrete representations learned through vector quantization. Theoretical support and empirical evidence demonstrate VQMoE's ability to overcome traditional router issues, showing a improvement in robustness over other SMoE routing methods while maintaining strong fine-tuning performance.

**Strengths:**

1. The introduction of Vector Quantization is interesting.

2. The writing is clear.

**Weaknesses:**

1. The definition of symbols is confusing, especially on Page 3.

For example, what does the v_k mean in Line 129?

What does the Eq. (3) mean in Line 132?

f^v in Line 142 is not present in Eq. (4).

What does the g(x)_c + g(x)_d mean in Line 148?

2. The lacked comparison.

In my understanding, this work maps samples into several discrete embeddings, i.e., the codebook. Thus, each sample corresponds to one embedding and use this embedding for routing. This is similar to [1]. [1] clusters samples into different cluster centers and generates one embedding for each cluster center. Then, [1] uses embedding for routing. Thus, can the authors provide a comparison with this method?

[1] Gou Y, Liu Z, Chen K, et al. Mixture of cluster-conditional lora experts for vision-language instruction tuning.

3. The additional training need for the pre-training stage.

In general, the MoE can be directly used to replace the FFN of the LLM in the fine-tuning stage. However, this work must train the codebook in the pre-training stage. Thus, I kind of disagree with the claim made in Line 367. Although not training the routing in fine-tuning, the additional training time is needed for pre-training.

Besides, I would like to know that whether this technique works for unbalanced datasets. When most samples correspond to one vector in the codebook, this technique may be hard to prevent the collapse. It will be helpful to provide some explanations about it.

In conclusion, although the writing is clear, the rigorousness should be enhanced. In addition, I am concerned about the lacked comparison and the additional training need for the pre-training. Thus, I vote for 5 now.

**Questions:**

Please refer to the Weaknesses.

---

> ### Author Response · Authors · 2024-11-17
> **Authors response to Reviewer YzaE**
>
> We sincerely appreciate the reviewer’s constructive suggestions and have provided our responses below.
>
> **W1. The definition of symbols is confusing, especially on Page 3.**
>
> A1.
>
> We apologize for the confusion caused by the symbols and have corrected it at our new revision. We have revised them as follows: $g(x)_c(x) = col_0(G(x))$, $g(x)_d(x) = col_1(G(x))$ is gating function for continuous and discrete representation with $G(x) = \operatorname{softmax}(W_g^T \times x)$. $W_g^T \in \mathbb{R}^{2 \times d}$ is a learnable weight.
>
> **W2. The lacked comparison. Can the authors provide a comparison with the MoCLE method?**
>
> A2.
>
> We experimented with a clustering-based approach similar to MoCLE[1], but found it unsuitable for our method. Compared with the MoCLE[1] approach, Vector Quantization provides the model with greater flexibility to learn cluster representations during training, making it more competitive than the clustering approach in practical applications. The training results on the Enwik8 dataset are presented in the Table below.
>
> | TopK | # Experts | SMoE | MoCLE | VQMoE | Metric |
> | -------| ------------ | -------- | -------- | -------- | --------- |
> | 1 | 16 | 1.28 | 1.29 | 1.25 | BPC |
> | 2 | 16 | 1.26 | 1.28 | 1.25 | BPC |
> | 4 | 16 | 1.26 | 1.28 | 1.25 | BPC |
> | 8 | 16 | 1.27 | 1.28 | 1.25 | BPC |
> | 16 | 16 | 1.27 | 1.28 | 1.25 | BPC |
>
> **W3. The additional training need for the pre-training stage.**
>
> A3.
>
> For the pretraining task, the continuous representation is learned using a top-1 router, while the discrete representation is learned through vector quantization. As a result, the total training resources are equivalent to those required for training a Top-2 router, as in [2], [3]. Therefore, no additional training is needed during the pretraining stage.
>
> **W4. This work must train the codebook in the pre-training stage. Thus, I kind of disagree with the claim made in Line 367. Although not training the routing in fine-tuning, the additional training time is needed for pre-training.**
>
> A4.
>
> As stated in Theorem 4.1 of our paper, the conventional MoE approach, which selects experts based on a router, can lead to inconsistent expert selections. To address this, we propose selecting experts by learning a discrete representation and assigning each input to its corresponding expert. For the pretraining task, the total computational resources remain unchanged; however, our method saves 28% of computational resources when applied to downstream tasks.
>
> **W5. Besides, I would like to know that whether this technique works for unbalanced datasets. When most samples correspond to one vector in the codebook, this technique may be hard to prevent the collapse. It will be helpful to provide some explanations about it.**
>
> A5.
>
> Yes, our method can handle unbalanced datasets. However, we need to employ some collapse-prevention techniques from the VQVAE field, such as using a frozen codebook and implicitly generating codes through a linear projection [4] or Online clustered codebooks as [5].
>
>
> **Reference:**
>
> [1] Gou, Y., Liu, Z., Chen, K., Hong, L., Xu, H., Li, A., Yeung, D., Kwok, J., & Zhang, Y. (2024) Mixture of cluster-conditional lora experts for vision-language instruction tuning. https://arxiv.org/abs/2312.12379
>
> [2] Chi, Z., Dong, L., Huang, S., Dai, D., Ma, S., Patra, B., Singhal, S., Bajaj, P., Song, X., Mao, X., Huang, H., & Furu Wei, F. (2022) On the representation collapse of sparse mixture of experts https://arxiv.org/abs/2204.09179
>
> [3] Dai, D., Dong, L., Ma, S., Zheng, B., Sui, Z., Chang, B., & Wei, F. (2022) Stablemoe: Stable routing strategy for mixture of experts. https://arxiv.org/abs/2204.08396
>
> [4] Zhu, Y., Li, B., Xin, Y., & Xu, L. (2024) Addressing representation collapse in vector quantized models with one linear layer. https://arxiv.org/abs/2411.02038
>
> [5] Zheng, C., & Vedaldi, A. (2023) Online clustered codebook. https://arxiv.org/abs/2307.15139

---

> > ### Author Response · Authors · 2024-11-22
> >
> > Dear Reviewer YzaE,
> >
> > Thank you for your constructive review. If you have any remaining questions or concerns, please feel free to let us know so we can address them before the deadline. If your concerns have been resolved, we kindly ask that you update your evaluation accordingly.
> >
> > Thank you very much!

---

> > > ### Comment · Reviewer_YzaE · 2024-11-25
> > >
> > > Thank you for the author's rebuttal.
> > > Sorry for the late response.
> > >
> > > I still have some concerns about W2 and W5.
> > > Regarding W2, what is the specific configuration of the experiment? Why can't I find relevant data from the manuscript, such as 1.25 for VQMoE? I am also curious why the performance of Top16 and Top1 is similar.
> > > Regarding W5, it intuitively makes sense to use a frozen codebook, but it would be even better to have some experimental examples.
> > >
> > > Taking into account the comments from other reviewers as well as the concerns that have not been fully addressed, I keep my score unchanged now.

---

> ### Author Response · Authors · 2024-11-25
> **[P2] Authors response to Reviewer YzaE**
>
> Dear Reviewer YzaE,
>
> Thank you for your reponse. We would like to address your concen as below
>
> **C1.  Regarding W2, what is the specific configuration of the experiment? Why can't I find relevant data from the manuscript, such as 1.25 for VQMoE**
>
> A1.
>
> For the configuration, we trained Transformer-XL at the base scale for **50k** steps across **VQMoE, SMoE, and MoCLE** to ensure a fair comparison within the limited rebuttal time. Since VQMoE learns a discrete representation where each cluster corresponds directly to an expert, its results for every $K$ are equal to $top1$: **1.25**.
>
> **C2.  this technique works for unbalanced datasets**
>
> A2.
>
> **Unbalanced datasets are not the focus of our research.** Our paper has demonstrated efficiency in the general setting through both theoretical analysis and experiments. Addressing unbalanced datasets is planned for future research applications.
>
>
> **Could you kindly point out any concerns that have not been fully addressed? If you have any further questions or issues, we would be happy to discuss them.**

---

### Official Review · Reviewer_gqJv · 2024-11-03

**Soundness:** 1
**Presentation:** 1
**Contribution:** 1
**Rating:** 3
**Confidence:** 3

**Summary:**

This paper presents VQ-MoE approach that uses both sparse MoE and a MoE that is routed based on some vector quantization of the tokens. It claims to theoretically demonstrate that this discrete representation is optimal and addresses the issue of collapse. It presents results in text and image.

**Strengths:**

I did not understand this work to be able to identify strengths.

**Weaknesses:**

Very confusingly written and not a clear explanation of the method. Despite reading this work on 3 different occasions I still do not understand the method. I am not able to explain what it does despite being familiar with both MoE and VQ-VAE. In some handwaving way it seems it pre-trains both a sparse moe and a “table lookup” based on a quantization of the intermediate representations and then it discards the sparse moe and only uses the “vq” table lookup during finetune.

I am not sure the scale of the data and experiments are enough to draw any conclusions. For example most MoE papers pretrain on a very different scale of thing. For image results SoftMoe pretrained on JFT-4B and shows zero-shot numbers in CIFAR-100 (ViT-B/16 - 71.0, Soft MoE-S/16 -  67.2) significantly above the **finetune** numbers shown here (67.3 VQMoE) . I think overall the scale of examples here are just not suitable for exploring and comparing MoE methods.

XMoE shows wikitext-103 results where a dense transformer reaches a pplx (22.61) smaller than any reported by this work (23.85 for large VQMoE).

**Questions:**

Please provide clear explanations of what is done at pre-training, finetune and inference so one can clearly see what it attempts to be done.

What is the setup for image tasks? What was the model pretrained on?

What does your dense baseline obtains in wikitext-103 and why is it so different from XMoE reported numbers?

---

> ### Author Response · Authors · 2024-11-17
> **Authors response to Reviewer gqJv**
>
> Thank you for your review. We hope our response below fully addresses your concerns and questions. Please let us know if you have any additional questions.
>
> **W1. Very confusingly written and not a clear explanation of the method.**
>
> A1.
>
> - We theoretically demonstrate that learning a discrete representation is a more optimal approach for expert selection compared to relying on a router. However, relying solely on training a discrete representation may lead to underfitting when dealing with large-scale data in practice.
>  - For the pretraining task, we propose simultaneously learning both discrete and continuous representations. The continuous representation is learned using a top-1 router, while the discrete representation is learned through vector quantization.
> - For the downstream tasks, we propose fine-tuning the pretraining model using only the discrete representation, which achieves better performance and improves speed by over 28%.
>
> **W2. Not sure the scale of the data and experiments are enough to draw any conclusions.**
>
> A2.
>
>  - While we agree with your concerns, scaling up to billion-scale models is not the primary focus of our work due to resource limitations. Instead, our contribution lies in proposing a discrete representation mechanism that optimally routes tokens to experts and proves effective for downstream tasks.
> -  For language tasks, we progressively scale the model from a few million parameters to several hundred million parameters and consistently observe that our method outperforms existing approaches. For vision tasks, we demonstrate the effectiveness of our method across various classification datasets. We are currently conducting additional experiments on the ImageNet dataset and will share the results with you once available.
>
> **Q1.  Please provide clear explanations of what is done at pre-training, finetune and inference so one can clearly see what it attempts to be done?**
>
> A3.
>
> - As illustrated in Figure 1 and detailed in Equations 4 and 5, we propose a pretraining approach that simultaneously learns both discrete and continuous representations. The continuous representation is obtained using a Top-1 Router, while the discrete representation is learned via vector quantization.
> - For the downstream tasks using the pretraining model, we propose fine-tuning the pretraining model using only the discrete representation, which achieves better performance and speed.
>
> ***Q2. What is the setup for image tasks? What was the model pretrained on?**
>
> A4.
>
> For image tasks, we conduct experiments on training image classification models from scratch across various datasets. Our training setup is based on the repository available at:  https://github.com/horrible-dong/QTClassification.
>
> **W3. XMoE shows wikitext-103 results where a dense transformer reaches a pplx (22.61) smaller than any reported by this work (23.85 for large VQMoE)**
>
> **Q3. What does your dense baseline obtains in wikitext-103 and why is it so different from XMoE reported numbers?**
>
> A5.
>
> - Are you referring to 22.61 PPL on the OpenWebText that is reported by Yange et al.  2024[1] ?. The model XMoE which we mentioned in our work is proposed by Chi et al. 2022[2].
> - The study by [2]  trained their model with more than twice the number of parameters and significantly more iterations compared to our approach, leading to results that differ from those presented in our paper.
>
> [1] Yang, Y., Qi, S., Gu, W., Wang, C., Gao, C., & Xu, Z. (2024) Xmoe: Sparse models with fine-grained and adaptive expert selection. https://arxiv.org/abs/2403.18926
>
> [2] Chi, Z., Dong, L., Huang, S., Dai, D., Ma, S., Patra, B., Singhal, S., Bajaj, P., Song, X., Mao, X., Huang, H., & Furu Wei, F. (2022) On the representation collapse of sparse mixture of experts https://arxiv.org/abs/2204.09179

---

> > ### Author Response · Authors · 2024-11-22
> >
> > Dear Reviewer gqJv,
> >
> > Thank you for your review. If you have any remaining questions or concerns, please let us know so we can address them before the deadline. If your concerns have been resolved, we kindly ask you to update your evaluation accordingly.
> >
> > Thank you so much!

---

> > > ### Author Response · Authors · 2024-11-25
> > > **Kindly Request for Reviewer's Feedback**
> > >
> > > Dear reviewer gqJv,
> > >
> > > Thank you for dedicating your time and effort to provide a review of our work. As the discussion period nears its conclusion, we hope you’ve had the opportunity to review our rebuttal. We believe our response has helped clarify key aspects of our work. If our rebuttal has satisfactorily addressed your concerns, we kindly request you to consider updating your review to reflect this. Of course, we remain open to further discussion should you have any additional questions or feedback.

---

### Official Review · Reviewer_ckSP · 2024-11-04

**Soundness:** 4
**Presentation:** 3
**Contribution:** 3
**Rating:** 5
**Confidence:** 3

**Summary:**

This submission proposes VQMoE, a new variant for sparse mixture of expert models (SMoE).
VQMoE utilizes the vector-quantization technique to construct a router, which is conventionally
made of softmax-based subnetwork and tends to collapse.
VQMoE adopts two-stage training: in the first stage, the VQ and conventional router are parallelly trained,
and in the second stage, the conventional router is removed and routing is solely performed by VQ.
Theoretical analyses claim that VQMoE performs consistent expert selection in terms of distance metric in the latent space,
and VQMoE's expert selection strategy (simply assigning a expert to each cluster) is optimal.
Experiments are conducted in language modelling, down stream tasks, and computer-vision tasks. and they demonstrate
better parameter-performance trade-off by VQMoE than existing MoE variants.

**Strengths:**

- The idea of performing routing by clustering is intuitive and promising.
- The proposed method is simple and seemingly easy to implement, which shows the potential of the proposed method for replacing existing MoE-based methods.
- The experiments are extensively conducted in language and vision domain.

**Weaknesses:**

- Proposition 4.2:
I have concerns in this statement’s mathematical rigorousness.
Optimality in what metrics is discussed in the proposition? Training error or loss?
What assumption is used in the proof?
It looks that $E_j(x_j) = \min_{c \in [1,k]}(E_j(x_c))$ (l. 795) requires that expert E_j is optimally trained with sample x_j.
Then, the proposition is claiming "if VQMoE is optimally trained with expert assignment strategy to asign E_j to x_j, samely assigning E_j to x_j is optimal in inference"
is nearly a tautology.

- On the two-stage training:
The necessity of the two-stage training is not experimentally confirmed.
1) How good is the first-stage-only training (i.e., VQ loss is used as a regularizer but the conventional Top-k routers are kept and used in inference) results?
2) Is it impossible to start the second-stage training (i.e,. train VQ routers without  Top-k routers from scratch) from scratch?

**Questions:**

Please see Weaknesses.

---

> ### Author Response · Authors · 2024-11-17
> **Author response to Reviewer ckSP**
>
> We are grateful for the reviewer’s constructive suggestions and provide our responses below.
>
> **W1. Proposition 4.2: I have concerns in this statement’s mathematical rigorousness. Optimality in what metrics is discussed in the proposition? Training error or loss? What assumption is used in the proof?**
>
> A1.
>
> - For optimality we discussed in Proposition 4.2 is an expert selection that minimizes training loss function.
> - About metrics or loss functions, it can be applied for any training loss function, for instance: Cross Entropy Loss or The negative log  likelihood loss.
> - About proof assumption, please refer to our above general response.
> - The question about optimal training selection vs optimal inference selection is out of scope of this Section 4.1 Theory and Section B proof. However, it is related to the generation theory of a mixture of experts, which was proved by [1] and [2].
>
> **W2. On the two-stage training: The necessity of the two-stage training is not experimentally confirmed.**
>
> A2.
>
> - For the question about two-stage training required, the answer is No. Table 1 and Table 3 we report the training model from scratch and our method outperforms the existing baseline.
> - Besides that, we finger out that the discrete representation that we learning from massive of data can be useful for a downstream task which was illustrated as the fine-tuning results at Table 2.
>
> **Q1. How good is the first-stage-only training (i.e., VQ loss is used as a regularizer but the conventional Top-k routers are kept and used in inference) results?**
>
> A3.
>
>  - For the pretraining task, we utilize both Top-1 Router for continuous representation learning and VQ for learning discrete presentation.
>  - For the finetuning task, we excluded Top-1 Router for both finetuning and inference. And we showed that discrete presentation is efficient in terms of both performance and inference speed.
>
> **Q2. Is it impossible to start the second-stage training (i.e,. train VQ routers without Top-k routers from scratch) from scratch?**
>
> A4.
>
> The answer is yes, however, training the discrete representation only approach might only work for tiny/base scale with a medium dataset in practice. The below results illustrates that point with our experiments as a below Table results on the Enwik8 dataset.
>
> Scale | TopK | # Experts | SMoE | VQMoE (Discrete Only) |
> |:------: |:------: |:------: |:------: |:------: |
> |    Tiny    | 1  | 16 | 1.28 | 1.25 |
> |    Tiny   | 2  | 16 | 1.26 | 1.25 |
> |    Tiny     | 4  | 16 | 1.26 | 1.25 |
> |     Tiny    | 8  | 16 | 1.27 | 1.25 |
> |     Tiny   | 16  | 16 | 1.27 | 1.25 |
> |   | | | | |
> |    Base    | 1  | 16 | 1.22 | 1.18 |
> |    Base   | 2  | 16 | 1.20 | 1.18 |
> |    Base     | 4  | 16 | 1.21 | 1.18 |
> |     Base    | 8  | 16 | 1.21 | 1.18 |
> |     Base   | 16  | 16 | 1.21 | 1.18 |
> |   | | | | |
> |    Large    | 1  | 64 | 1.12 | 1.14 |
> |    Large   | 2  | 64 | 1.09 | 1.14 |
> |    Large     | 4  | 64 | 1.09 | 1.14 |
> |     Large    | 8  | 64 | 1.09 | 1.14 |
> |     Large   | 16  | 64 | 1.10 | 1.14 |
> |     Large   | 16  | 64 | 1.10 | 1.14 |
> |     Large   | 16  | 64 | 1.12 | 1.14 |
> |   | | | | |
>
> [1] Chen, Z., Deng, Y., Wu, Y., Gu, Q., & Li. Y. (2022) Towards understanding mixture of experts in deep learning. https://arxiv.org/abs/2208.02813
>
> [2] Zhao, J., Wang, P., & Wang. Z. (2024) Generalization error analysis for sparse mixture-of-experts: A preliminary study.  https://arxiv.org/abs/2403.17404

---

> > ### Author Response · Authors · 2024-11-22
> >
> > Dear Reviewer ckSP,
> >
> > Thank you for your thoughtful review. If you have any remaining questions or concerns, please let us know so we can address them before the deadline. If your concerns have been resolved, we kindly ask you to update your evaluation accordingly.
> >
> > Thank you again!

---

> > > ### Comment · Reviewer_ckSP · 2024-11-23
> > >
> > > I appreciate the Authors' answer.
> > >
> > > Unfortunately, my largest concern, W1, about Proposition 4.2, is not solved well.
> > > In my understanding, the proposition claims that "after experts were trained with expert-assignment strategy A,
> > > the expert-assignment strategy A is optimal in terms of training loss".
> > > However, this is likely to be true for any strategy, so I think that this can not highlight the characteristics of the proposed one.
> > > Claiming that "learning a discrete representation is an optimal approach for expert selection" on the basis of this may be misleading.
> > >
> > > I would keep my initial rating.

---

> ### Author Response · Authors · 2024-11-23
> **[P2] Authors response to Reviewer ckSP**
>
> Dear Reviewer ckSP,
>
> Thank you so much for your reponse. We would like to adress your concern about the **Proposition 4.2** as below:
>
> 1. Our idea for Proposition 4.2 is a optimal to training experts rather than selection by each training steps:
>
> - **At the first training iteration**, since all experts are initialized randomly, so any pair $(t_j, E_i)$ can be assigned, where $t_j \in C_j$ is a token belong to $j_{th}$ cluster, and $E_i$ is the $i_{th}$ expert.
> - **From the second training iteration onward**, the optimal strategy is to maintain the assignment $j_{th}$ cluster to $i_{th}$ expert. This is because the $i_{th}$ expert has already started learning knowledge specific to the $j_{th}$ cluster during the previous step, making it the most effective choice among $N$ possible experts (where $N$ is the total number of experts).
>
> 2. Additionally, we want to emphasize that the core theoretical contribution of our work is presented in Section 4.2, **"VQMOE Solves Representation Collapse by Design."** In this section, we demonstrate that our method surpasses other routing approaches in effectively addressing representation collapse. Consequently, our approach proves to be more efficient than alternative routing methods, as supported by empirical results as Section 5.
>
> If you have any further questions or concerns, we would be delighted to discuss them.
>
> Thank you so much.

---

### Author Response · Authors · 2024-11-13

We sincerely thank all reviewers for their time and effort in providing constructive feedback. We will respond to each question and comment as soon as possible.

---

> ### Author Response · Authors · 2024-11-17
> **Genenral Response**
>
> We would like to express our sincere gratitude to all the reviewers for their insightful comments and constructive feedback, which have greatly helped us refine and improve our approach. In the following response, we address each reviewer’s questions and concerns individually. Here, we provide a summary of the revisions made to the manuscript and respond to the common questions raised by multiple reviewers.
>
> **Summary of Paper Revisions**
> ---------------------------------------
>
> - Revised Formulas 4 and 5, as well as Figure 1, to improve clarity and accuracy.
> - Updated Subsection 4.1 (Theoretical Analysis) and Section B (Theoretical Proof) with enhanced mathematical notation and more detailed explanations.
> - Added further explanations to Figures 4 and 5 for better interpretability.
> - Corrected several writing errors for improved accuracy and coherence.
>
>
> **Responses to Common Questions**
> ----------------------------------------------
> **Q1. Is this method required additional training or two stage training ?**
>
> A1. No, our method requires only single-stage training, similar to a conventional SMoE. For the pretraining phase, we leverage both a Top-1 Router for continuous representation learning and a VQ module for learning discrete representations. This results in no additional computational overhead compared to a Top-2 Router MoE. When applying the pretrained model to downstream tasks, we propose excluding the Top-1 Router during both fine-tuning and inference, showing that discrete representations are efficient in terms of both performance and inference speed.
>
> **Q2. What is the assumption for the section 4.1 Theory Analysis and is this possible for real data?**
>
> A2. We build upon the assumptions of [1] and [2] that the data exhibits a clustering structure. Besides that, other works in MoE, such as [3] and [4], assume that the data contains specific patterns. From a theoretical perspective, [2] demonstrated that the Mixture of Experts (MoE) can be interpreted as a clustering problem. If the number of clusters is smaller than the number of experts, collapse may occur; conversely, if it is larger, the model could underfit. In practice, selecting an appropriate number of experts can be achieved through hyperparameter tuning tailored to the specific dataset.
>
>
> **Q3. Can this method scale up language models to billion parameters?**
>
> A3. Scaling up to billion-scale language models is not the primary focus of our work due to resource limitations. Instead, our contribution lies in proposing a discrete representation mechanism that optimally routes tokens to experts and proves effective for downstream tasks.
> For language tasks, we progressively scale the model from a few million parameters to several hundred million parameters and consistently observe that our method outperforms existing approaches. For vision tasks, we demonstrate the effectiveness of our method across various classification datasets. We are currently conducting additional experiments on the ImageNet dataset and will share the results with you once available.
>
> **Q4. Can this method work with a large scale vision dataset such as Imagenet?**
>
> A4. We are training the baselines and our method on ImageNet and we will report the result as soon as possible.
>
> **Reference:**
>
> [1] Chen, Z., Deng, Y., Wu, Y., Gu, Q., & Li. Y. (2022) Towards understanding mixture of experts in deep learning. https://arxiv.org/abs/2208.02813
>
> [2] Dikkala, N., Ghosh, N., Meka, R., Panigrahy, R., Vyas, N., & Wang. X. (2023) On the benefits of learning to route in mixture-of-experts models. https://aclanthology.org/2023.emnlp-main.583.pdf
>
> [3] Chowdhury, M., Zhang, S., Wang, M., Liu, S., & Chen. P. (2023) Patch-level routing in mixture-of-experts is provably sample-efficient for convolutional neural networks. https://arxiv.org/pdf/2306.04073
>
> [4] Chowdhury, M., Wang, M., Maghraoui, K., Wang, N., Chen, P., &Carothers. C. (2024) A
> provably effective method for pruning experts in fine-tuned sparse mixture-of-experts. https://arxiv.org/abs/2405.16646

---

> > ### Author Response · Authors · 2024-11-22
> > **Benchmark on a large scale vision dataset such as Imagenet**
> >
> > **Q4(continue). Can this method work with a large scale vision dataset such as Imagenet?**
> >
> > A4. Yes, VQMoE consistently outperforms the dense model and other routing methods on large-scale datasets such as ImageNet, across both small and large-scale versions. Please refer to the table below for more details.
> >
> > | Size | Data | ViT | SoftMoE | SMoE | StableMoE | XMoE | VQMoE |
> > |:--:|:--:| :---: | :---: | :---: | :---: | :---: | :---: |
> > |Small - 10M | ImageNet-1K | 52.2 | 41.6 | 52.8 | 52.5 | 53.5 | **54.8** |
> > |Large - 110M | ImageNet-1K | 71.1 | 68.2 | 71.0 | 70.6 | 70.8 | **71.3** |

---

### Meta-Review · Area_Chair_VMh5 · 2024-12-22

**Metareview:**

This paper proposes to mitigate the issues of routing inconsistency and representation collapse in Sparse Mixture of Experts models.  By using discrete representations learned via vector quantization, it improves the expert selection process, avoiding traditional router-based methods.

This paper has received 5 reviews, with final ratings 5,5,5,5,3.  The reviewers appreciated the novel vector quantization idea used in this setting, and the authors' extensive rebuttals with additional experimental results.  However, despite the rebuttal and clarifications, the consensus among the reviewers was that the paper had significant weaknesses, especially in terms of theoretical rigor, clarity of presentation, marginal performance gains, and the lack of large-scale experiments.  The final recommendation is rejection based on the overall unconvincing experimental validation and unresolved theoretical issues.

**Additional Comments On Reviewer Discussion:**

Reviewers appreciated the authors' extensive rebuttal, but their main concerns largely remained.

---

### Decision · Program_Chairs · 2025-01-22

Reject